# RNA polymerase errors cause splicing defects and can be regulated by differential expression of RNA polymerase subunits

Lucas B Carey*

Department of Experimental and Health Sciences, Universitat Pompeu Fabra, Barcelona, Spain

**Abstract** Errors during transcription may play an important role in determining cellular phenotypes: the RNA polymerase error rate is >4 orders of magnitude higher than that of DNA polymerase and errors are amplified >1000-fold due to translation. However, current methods to measure RNA polymerase fidelity are low-throughout, technically challenging, and organism specific. Here I show that changes in RNA polymerase fidelity can be measured using standard RNA sequencing protocols. I find that RNA polymerase is error-prone, and these errors can result in splicing defects. Furthermore, I find that differential expression of RNA polymerase subunits causes changes in RNA polymerase fidelity, and that coding sequences may have evolved to minimize the effect of these errors. These results suggest that errors caused by RNA polymerase may be a major source of stochastic variability at the level of single cells.

*For correspondence: lucas. carey@upf.edu

**Competing interests:** The author declares that no competing interests exist.

The information that determines protein sequence is stored in the genome, but that information must be transcribed by RNA polymerase and translated by the ribosome before reaching its final form. DNA polymerase error rates have been well characterized in a variety of species and environmental conditions, and are low – on the order of one mutation per $10^8$–$10^{10}$ bases per generation (*Lynch, 2011*; *Lang and Murray, 2008*; *Zhu et al., 2014*). In contrast, RNA polymerase errors are uniquely positioned to generate phenotypic diversity. Error rates are high ($10^{-6}$–$10^{-5}$) (*Gout et al., 2013*; *Lynch, 2010*; *Shaw et al., 2002*; *de Mercoyrol et al., 1992*), and each mRNA molecule is translated into 2000–4000 molecules of protein (*Schwanhäusser et al., 2011*; *Futcher et al., 1999*), resulting in the amplification of any errors. Likewise, because many RNAs are present at an average of less than one molecule per cell in microbes (*Pelechano et al., 2010*) and in embryonic stem cells (*Islam et al., 2011*), an RNA with an error may be the only RNA for that gene; all newly translated protein will contain this error. Despite the fact that transient errors can result in altered phenotypes (*Gordon et al., 2013, 2015*), the genetics and environmental factors that affect RNA polymerase fidelity are poorly understood. This is because current methods for measuring polymerase fidelity are technically challenging (*Gout et al., 2013*), require specialized organism-specific genetic constructs (*Irvin et al., 2014*), and can only measure error rates at specific loci (*Imashimizu et al., 2013*).

To overcome these obstacles I developed MORPhEUS (Measurement Of RNA Polymerase Errors Using Sequencing), which enables measurement of differential RNA polymerase fidelity using existing RNA-seq data (*Figure 1*). The input is a set of RNA-seq fastq files and a reference genome, and the output is the error rate at each position in the genome. I find that RNA polymerase errors result in intron retention and that cellular mRNA quality control may reduce the effective RNA polymerase error rate. Moreover, my analyses suggest that the expression level of the RPB9 Pol II

**eLife digest** Genes encode instructions to make proteins and other molecules. To issue an instruction, a gene is first used as a template to make molecules of ribonucleic acid (called mRNAs for short) in a process called transcription. An enzyme called RNA polymerase – which comprises several protein subunits that all work together – is responsible for making the mRNA molecules. Occasionally, this enzyme makes mistakes that lead to small changes in the instruction that is produced. These mistakes are rare, but because cells make thousands of mRNAs, a single human cell can make 10-100 transcription errors per second.

It has been difficult to study how often RNA polymerase makes mistakes and what effect these mistakes have on organisms because the techniques available for research are labour-intensive and technically challenging. Here, Lucas Carey demonstrates that it is possible to use a technique called RNA sequencing to study the accuracy of RNA polymerase in human and yeast cells.

The experiments show that altering the levels of the different subunits of RNA polymerase in cells can change how many mistakes are made during transcription. This suggests that cells may be able regulate number of mistakes by controlling the production of specific subunits. Carey found that the severity of the mistakes made by RNA polymerase depends on where the mistake is in the mRNA. For example, errors in specific parts of the mRNA can alter how the whole instruction is edited later, while others might make only a tiny change to the protein encoded by the gene. Carey also found evidence that the instructions encoded by genes may have evolved in such a way to minimise the effect of any errors on their roles in cells.

RNA sequencing is less labour-intensive than other methods used to study the accuracy of RNA polymerase and is already used to address other research questions on a wide variety of different organisms. Therefore, Carey's findings will make it easier to study what genes or environmental factors influence the number of errors made during transcription. A major challenge for the future is to find out if the mistakes made by RNA polymerase can lead to cancer and other human diseases.

subunits Rpb9 and Dst1 (TFIIS) determines RNA polymerase fidelity in vivo. Because it can be run on any existing RNA-seq data, MORPhEUS enables the exploration of a previously unexplored source of biological diversity in microbes and mammals.

Technical errors from reverse transcription and sequencing, and biological errors from RNA polymerase look identical (single-nucleotide differences from the reference genome). Therefore, a major challenge in identifying single-nucleotide polymorphisms (SNPs) and in measuring changes in polymerase fidelity is the reduction of technical errors (*Kleinman and Majewski, 2012*; *Pickrell et al., 2012*; *Li et al., 2011*) (*Figure 1*). First, I map full-length (untrimmed) reads to the genome and discard reads with indels, with more than two mismatches, that map to multiple locations in the genome, and that do not map end to end along the full length of the read. Next, I trim the ends of the mapped reads, as alignments are of lower quality along the ends, and the mismatch rate is higher, especially at splice junctions. I also discard any cycles within the run with abnormally high error rates, and bases with low Illumina quality scores (*Figure 1—figure supplement 1*). Finally, using the remaining bases, I count the number of matches and mismatches to the reference genome at each position in the genome. I discard positions with identical mismatches that are present more than once, as these are likely due to subclonal DNA polymorphisms or sequences that Illumina miscalls in a systematic manner (*Meacham et al., 2011*) (*Figure 1—figure supplement 2*). The result is a set of mismatches, many of which are technical errors and some of which are RNA polymerase errors. In order to determine if RNA-seq mismatches are due to RNA polymerase errors, it is necessary to identify sequence locations in which RNA polymerase errors are expected to have a measurable effect, or situations in which RNA polymerase fidelity is expected to vary.

I reasoned that RNA polymerase errors that alter positions necessary for splicing should result in intron retention, while sequencing errors should not affect the final structure of the mRNA (*Figure 2a*). However, mutations in the donor and acceptor splice sites also result in decreased expression (*Jung et al., 2015*), and therefore are difficult to measure using RNA-seq. Therefore, I used chromatin-associated and nuclear RNA from Hela and Huh7 cells (*Dhir et al., 2015*), and extracted all reads that span an exon–intron junction for introns with canonical GT and AG splice

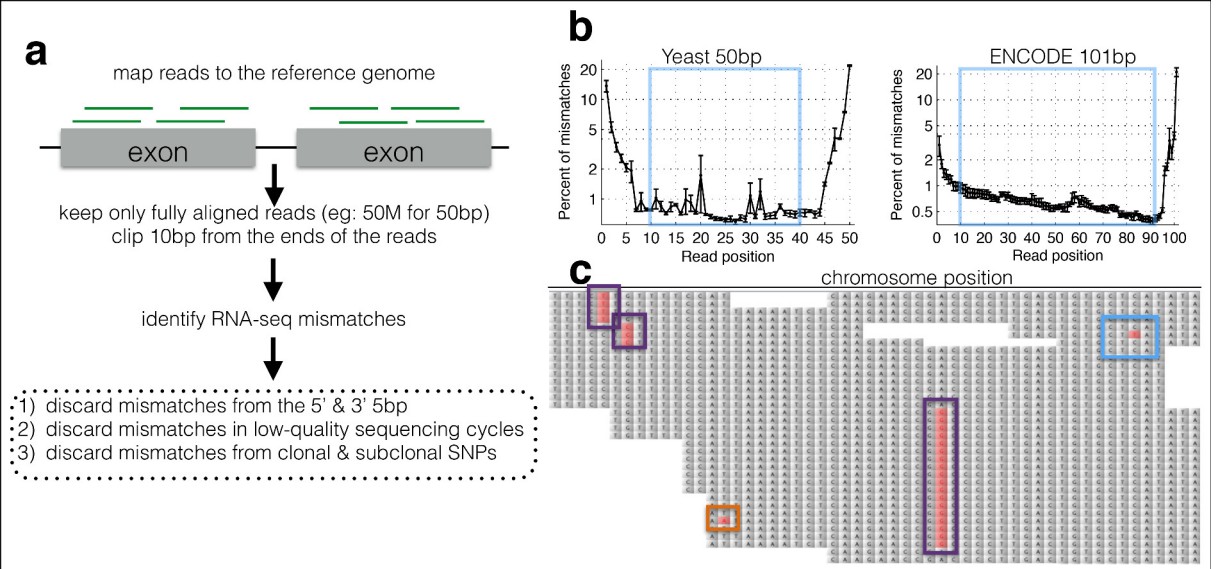

**Figure 1.** A computational framework to measure relative changes in RNA polymerase fidelity. (a) Pipeline to identify potential RNA polymerase errors in RNA-seq data. High quality full-length RNA-seq reads are mapped to the reference genome or transcriptome using bwa, and only reads that map completely with two or fewer mismatches are kept. (b) Then 10 bp from the front and 10 bp from the end of the read are discarded as these regions have high error rates and are prone to poor quality local alignments. (c) Errors that occur multiple times (purple boxes) are discarded, as these are likely due to subclonal DNA mutations or motifs that sequence poorly on the HiSeq. Unique errors in the middle of reads (cyan box) are kept and counted.
The following figure supplements are available for Figure 1:

**Figure supplement 1.** Cycle-specific error rates and better differentiation of genetically determined error rates using base quality value cutoffs.

**Figure supplement 2.** RNA-seq data are enriched for mismatches to the reference genome that occur far more often than expected.

sites, and measured the RNA-seq mismatch rate at each position. I find that errors at the G and U in the 5' donor site and at the A in the acceptor site are significantly enriched relative to errors at other positions (*Figure 2b*), and to errors in exonic trinucleotides at splicing motifs in the human genome (*Figure 2—figure supplement 1*) suggesting that RNA polymerase mismatches can result in changes in transcript isoforms. The ability of RNA polymerase errors to significantly affect splicing has been proposed (*Fox-Walsh and Hertel, 2009*) but never previously measured.

RPB9 is known to be involved in RNA polymerase fidelity in vitro and in vivo (*Irvin et al., 2014*; *Knippa and Peterson, 2013*). Therefore, I reasoned that cell lines expressing low levels of RPB9 would have higher RNA polymerase error rates. Consistent with this, I find that RPB9 expression varies eightfold across the ENCODE cell lines, and this expression variation is correlated with the RNA-seq error rate (*Figure 2c*, *Figure 2—figure supplement 2*). This suggests that low RPB9 expression may cause decreased polymerase fidelity in vivo.

In addition, export of mRNAs from the nucleus involves a quality-control mechanism that checks if mRNAs are fully spliced and have properly formed 5' and 3' ends (*Lykke-Andersen, 2001*). I hypothesized that mRNA export may involve a quality control that removes mRNAs with errors. I used the ENCODE dataset in which nuclear and cytoplasmic poly-A + mRNAs were sequenced; thus I can compare nuclear and cytoplasmic fractions from the same cell line grown in the same conditions and processed in the same manner. I find that the nuclear fraction has a higher RNA polymerase error rate than does the cytoplasmic fraction (*Figure 2c,d*), suggesting that either that nuclear RNA-seq has a higher technical error rate or that the cell has mechanisms for reducing the effective polymerase error rate by preventing the export of mRNAs that contain errors.

Rpb9 and Dst1 are known to be involved in RNA polymerase fidelity in vitro, yet there is conflicting evidence as to the role of Dst1 in vivo(*Shaw et al., 2002*; *Irvin et al., 2014*; *Knippa and Peterson, 2013*; *Nesser et al., 2006*; *Walmacq et al., 2009*; *Kireeva et al., 2008*). Part of these conflicts may result from the fact that the only available assays for RNA polymerase fidelity are special reporter

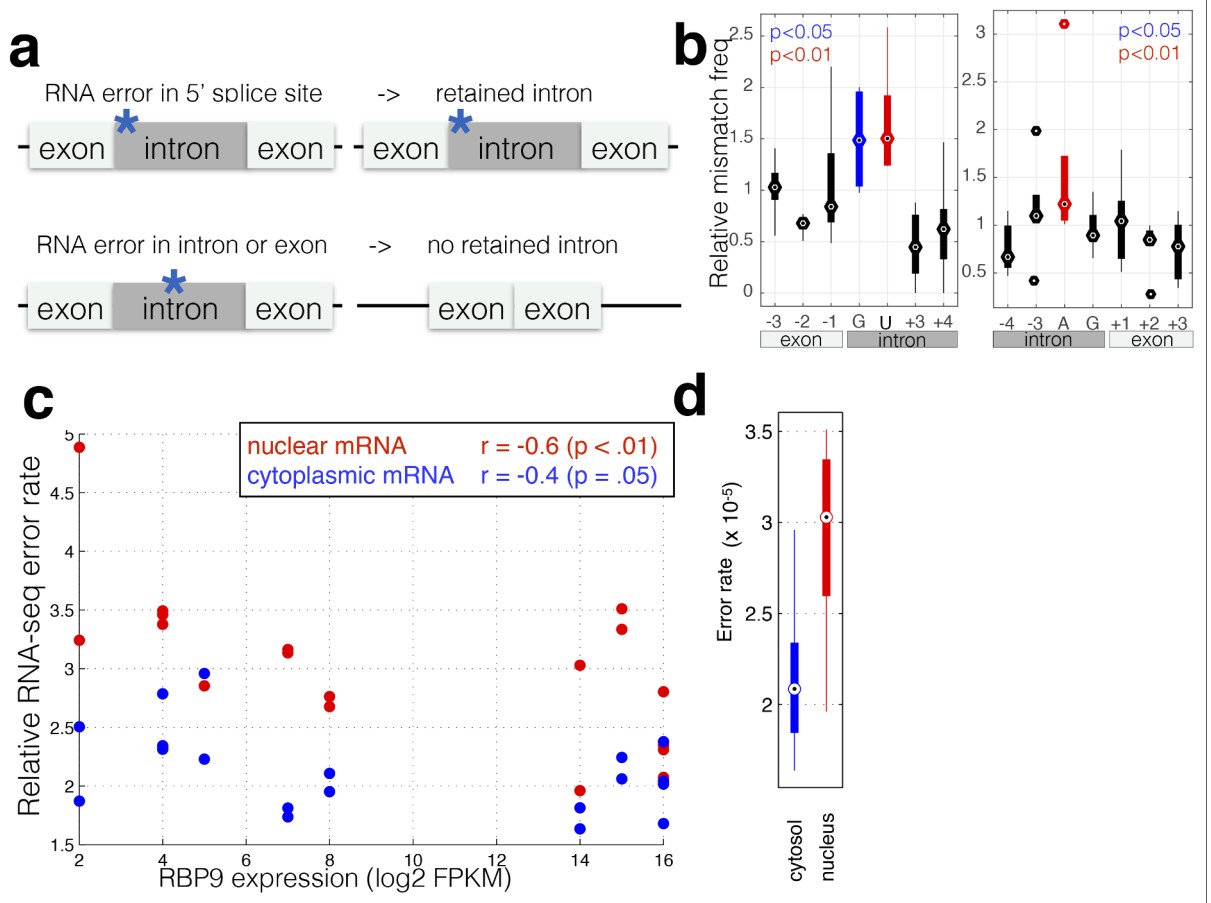

**Figure 2.** RNA polymerase errors cause intron retention and error rates are correlated with RPB9 expression. (a) RNA polymerase errors at the splice junction should result in intron retention, as DNA mutations at the 5' donor site are known to cause intron retention. (b) Shown are the RNA-seq mismatch rates at each position relative to the 5' donor splice site, for sequencing reads that span an exon–intron junction. Mismatch rates from chromatin-associated and nuclear RNAs are higher at the 5' and 3' splice sites, suggesting that RNA polymerase errors at this site result in intron retention. (c) For all ENCODE cell lines, RPB9 expression was determined from whole-cell RNA-seq data, and the RNA-seq error rate was measured separately for the cytoplasmic and nuclear fractions. (d) The RNA-seq error rate is higher (paired t-test, p=0.0019) in the nuclear than the cytoplasmic fraction, suggesting that quality-control mechanism may block nuclear export of low quality mRNAs.

The following figure supplements are available for Figure 2:

**Figure supplement 1.** RNA-seq mismatch rates for all trinucleotides in chromatin-associated and nuclear RNAs.

**Figure supplement 2.** RBP9 expression negatively correlates with RNA-seq mismatch rates.

strains that rely on DNA sequences known to increase the frequency of RNA polymerase errors. While I found that RPB9 expression correlates with RNA-seq error rates in mammalian cells, correlation is not causation. Furthermore, differences in RNA levels do not necessitate differences in stoichiometry among the subunits in active Pol II complexes. In order to determine if differential expression of RPB9 or DST1 are causative for differences in RNA polymerase fidelity in vivo, I constructed two yeast strains in which I can alter the expression of either RPB9 or DST1 using β-estradiol and a synthetic transcription factor that has no effect on growth rate or the expression of any other genes (*Mcisaac et al., 2014, 2013*). I grew these two strains (Z3EVpr-RPB9 and Z3EVpr-DST1) in different concentrations of β-estradiol and performed RNA-seq. I find that cells expressing low levels of RPB9 have high RNA polymerase error rates (*Figure 3a*). Likewise, cells with low DST1 have high error rates (*Figure 3a*). The increase in errors rate is not a property of cells defective for transcription elongation (*Figure 3—figure supplement 1*). The increase in error rates due to mutations in Rpb9 and Dst1 have not been

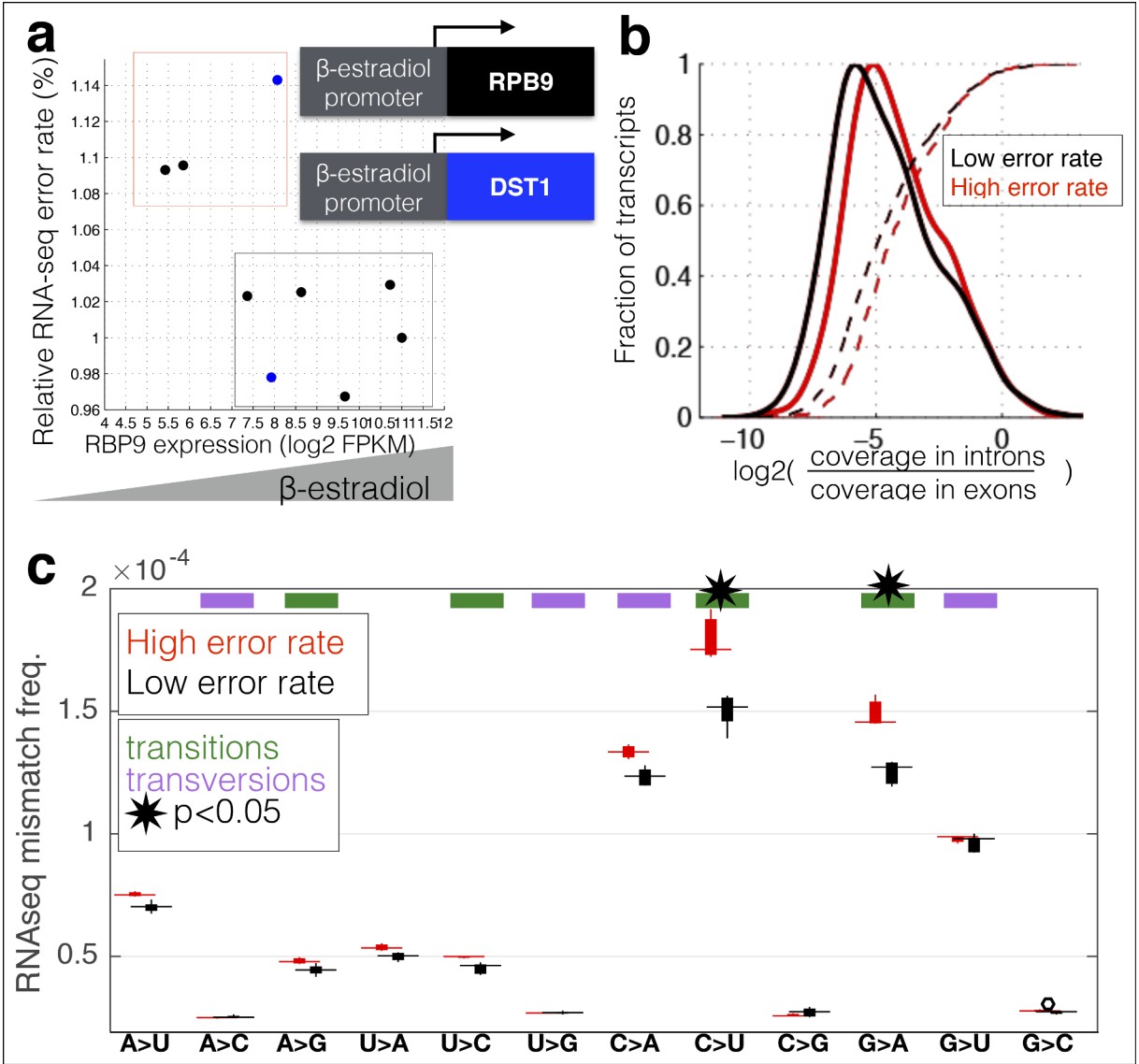

**Figure 3.** RNA polymerase error rate is determined by the expression level of RPB9 and DST1. (a) RNA-seq error rates I re-measured for two strains ($Z_3$EVpr-RPB9, black points, $Z_3$EVpr-DST1, blue points) grown at different concentrations of β-estradiol. The points show the relationship between RPB9 expression levels (determined by RNA-seq) and RNA-seq error rates. The blue points show RPB9 expression levels for the $Z_3$EVpr-DST1 strain, in which DST1 expression ranges from 16 fragments per kilobase per million (FPKM) at 0 nM β-estradiol to 120 FPKM native expression to 756 FPKM at 25 nM β-estradiol. Low induction of both DST1 or RPB9 results in high RNA-seq error rates (red box), while wild-type and higher induction levels result in low RNA-seq error rates (black box). (b) Across all genes, the intron retention rate is higher in conditions with low RNA polymerase fidelity (t-test between high and low error rate samples, p=0.029), consistent with the hypothesis that RNA polymerase errors result in splicing defects. (c) The error rate for each of the 12 single base changes are shown for induction experiments that gave high (red) or low (black) RNA-seq error rates. Transitions (G<–>A, C<–>U) are marked with green boxes and transversions (A<–>C, G<–>U) with purple.

The following figure supplements are available for Figure 3:

**Figure supplement 1.** Mutations that affect transcription elongation do not affect measured RNA-seq mismatch frequencies.

**Figure supplement 2.** Decreases in RPB9 and DST1 expression in yeast results in more single base insertions in RNA-seq data.

robustly measured, however, there are some rough numbers. Here, the measured increase in error rate is 13%, while the measured effect of Rpb9 deletion in vitro is fivefold (*Walmacq et al., 2009*) and in vivo following reverse transcription is 30% (*Nesser et al., 2006*). If 2% of the observed mismatches

are due to RNA polymerase errors, a fivefold increase in polymerase error rate results in a 10% increase in measured mismatch frequency; this is consistent with RNA polymerase fidelity of $10^{-6}$–$10^{-5}$ and overall RNA-seq error rates of $10^{-4}$. Note that in our assay cells still express low levels or RPB9, and we therefore expect the increase in error rate to be lower, suggesting that RNA polymerase errors constitute 5–10% of the measured mismatches. Our ability to genetically control the expression of DST1 and RPB9, and measure changes in RNA-seq error rates is consistent with MORPhEUS measuring RNA polymerase fidelity. In addition, we observe more single-nucleotide insertions in the RNA-seq data from the high error rate samples, suggesting that depletion of RPB9 and DST1 results in increased insertions in transcripts, but not increased deletions (*Figure 3—figure supplement 2*). Finally, genetic reduction in RNA polymerase fidelity results in increased intron retention, consistent with RNA polymerase errors causing reduced splicing efficiency (*Figure 3b*).

A unique advantage of MORPhEUS is that it measures thousands of RNA polymerase errors across the entire transcriptome in a single experiment, and thus enables he complete characterization of the mutation spectrum and biases of RNA polymerase. I asked how altered RPB9 and DST1 expression levels affect each type of single-nucleotide change. I find that, with decreasing polymerase fidelity, transitions increase more than transversions, and that C→U errors are the most common (*Figure 3c*). This result, along with other sequencing based results (*Gout et al., 2013*), have shown that DNA and RNA polymerase have broadly similar error profiles (*Zhu et al., 2014*); it will be interesting to see if all polymerases share the same mutation spectra, and if this is due to deamination of the template base, or is a structural property of the polymerase itself. Interestingly, I find that coding sequences have evolved so that errors are less likely to produce in-frame stop codons than out-of-frame stop codons, suggesting that natural selection may act to minimize the effect of polymerase errors (*Figure 4*).

Here I have presented proof that relative changes in RNA polymerase error rates can be measured using standard Illumina RNA-seq data. Consistent with previous work in vivo and in vitro, I find that depletion of RPB9 or Dst1 results in higher RNA polymerase error rates. Furthermore, I find that expression of RPB9 negatively correlates with RNA-seq error rates in human cell lines, suggesting that differential expression of RPB9 may regulate RNA polymerase fidelity in vivo in humans. In addition, consistent with the errors detected by MORPhEUS being due to RNA polymerase and not technical errors, in reads spanning an exon–intron junction, the measured error rate is higher at the 5' donor splice site, suggesting that RNA polymerase errors result in intron retention. Because it can be run on existing RNA-seq data, I expect MORPhEUS to enable many future discoveries regarding

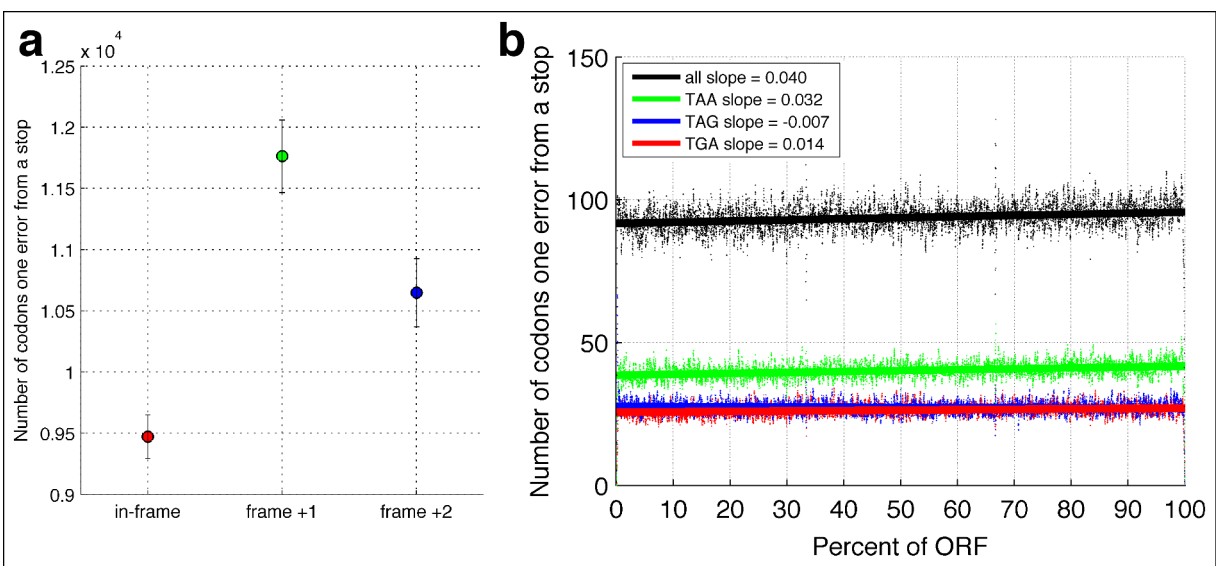

**Figure 4.** In-frame stop codons are less likely to be created by polymerase errors. For all genes in yeast, I calculated the number of codons which are one polymerase error from a stop codon. (a) Fewer in-frame codons can be turned into a stop codon by a single-nucleotide change, compared to out-of-frame codons. (b) Codons that are one error away from generating an in-frame stop codon are more likely to be found at the ends of the open reading frames (ORFs), compared to the beginning of the ORF.

both the molecular determinants of RNA polymerase error rates and the relationship between RNA polymerase fidelity and phenotype.

## Materials and methods

### Counting RNA polymerase errors in already aligned ENCODE data

Much existing RNA-seq data is available as bam files aligned to the human genome. In order to bypass alignment, which is the most computationally expensive step of the pipeline, I developed a method capable of using RNA-seq reads aligned with spliced aligners. First, in order to avoid increased mismatch rates at splice junctions due to alignment problems with both spliced and unspliced reads, I used SAMtools (*Li et al., 2009*) and awk to remove all alignments that do not align along the full length of the genome (e.g., for 76 bp reads, only reads with a CIGAR flag of 76 M). The remaining reads weretrimmed (bamUtil, trimBam) to convert the first and last 10 bp of each read to Ns and set the quality strings to '!'. I then used samtools mpileup (-q30 –C50 –Q30) and custom perl code to count the number of reads and number of errors at each position in genome. Positions with too many errors (e.g., more than one read of the same nonreference base) were not counted.

### Measurement of error rates at splice junctions

I used the University of California Santa Cruz (UCSC) table browser (*Karolchik, 2004*) to download two bed files: hg19 EnsemblGenes introns with -10 bp flanking from each side, and another file with the introns and +10 bp flanking on either side. I then used bedtools (*Quinlan and Hall, 2010*) (bedtools flank -b 20 -l 0 and bedtools flank -l 20 -b 0) to generate bed files with intervals that contain the splicing donor and acceptor sites, respectively. In addition, I used bedtools getfasta on the +10 bp flanking bed file to keep only introns flanked by GT and AG donor and acceptor sites. The final result is a pair of bam files with intervals centered on the splicing donor or acceptor sites. I used this new bed file to count error rates around each splice junction. The error rate at each position (e.g., -10, -9, -8, etc. from the G at the 5' donor site) is the sum of all errors at that position, divided by the sum of all reads. Positions are relative to the splicing feature, not to the genome, as error rates at any single genomic position are dominated by sampling bias. Per mono-, di-, and trinucleotide background error rates were-calculated using the same scripts, but without limiting mpileup to the splice junctions.

### Strain construction and RNA sequencing for RPB9 and DST1 strains

The parental strain DBY12394 (*Mcisaac et al., 2013*) (GAL2 + s288c repaired HAP1, ura3Δ, leu2Δ0:: ACT1pr-Z3EV-NatMX) was transformed with a polymerase chain reaction (PCR) product (KanMX-Z3EVpr) to generate a genomically integrated inducible RPB9 (LCY143) or DST1 (LCY142). To induce various levels of expression, strains were re-grown in YPD + 0-, 3-, 6-, 12-, or 25-nM β-estradiol (Sigma, St. Louis, MO, USA, E4389) for more than 12 hr to a final $OD_{600}$ of 0.1 – 0.4. Cellular RNA was extracted using the Epicenter MasterPure RNA Purification Kit, and Illumina sequencing libraries were prepared using the Truseq Stranded mRNA kit, and sequenced on an HiSeq2000 with at least 20,000,000 50 bp sequencing reads per sample.

I used bwa (*Li and Durbin, 2009*) (-n 2, to permit no more than two mismatches in a read) to align the yeast RNA-seq reads to the reference genome, and trimBam from bamUtil to mask the first and last 10 bp of each read. I used samtools mpileup (*Li et al., 2009*) (-q 30 -d 100000 -C50 –Q39) to count the number of reads and mismatches at each position in the genome, discarding low confidence mapping, reads that map to multiple positions, and low quality reads. Duplicate reads can be removed from the fastq file if the coverage is low enough so that all reads that map to identical genome coordinates are expected be PCR duplicates from the same RNA fragment. This is the case for low coverage paired-end reads with a variable insert size, but not for very high coverage datasets or single-ended reads.

### Pre-existing RNA-seq datasets

For the intron retention analysis in human cells, data are from NCBI SRA PRJNA253670. Data for the elc4 and spt4 analysis are from PRJNA167772 and PRJNA148851, respectively. For RPB9 correlation, undefined data (SRA PRJNA30709) are all from the Gingeras lab at CSHL.

## Acknowledgements

I thank members of the Carey lab and the computational genomics groups in the PRBB for thoughtful discussions.

## Additional information

### Funding

| Funder | Grant reference number | Author |
|---|---|---|
| Agència de Gestió d'Ajuts Universitaris i de Recerca | 2014 SGR 0974 | Lucas B Carey |

The funders had no role in study design, data collection and interpretation, or the decision to submit the work for publication.

### Author contributions

LBC, Conception and design, Acquisition of data, Analysis and interpretation of data, Drafting or revising the article, Contributed unpublished essential data or reagents

## Additional files

### Major datasets

The following datasets were generated:

| Author(s) | Year | Dataset title | Dataset ID and/or URL | Database, license, and accessibility information |
|---|---|---|---|---|
| Carey LB | 2015 | PRJNA289596 | http://www.ncbi.nlm.nih.gov/bioproject/289596 | Publicly available at the NCBI BioProject database (Accession no: PRJNA289596). |

The following previously published dataset was used:

| Author(s) | Year | Dataset title | Dataset ID and/or URL | Database, license, and accessibility information |
|---|---|---|---|---|
| The ENCODE Consortium | 2008 | Home sapiens (human) | http://www.ncbi.nlm.nih.gov/bioproject/30709 | Publicly available at the NCBI BioProject database (Accession no: PRJNA30709) |

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
