## [Decision Letter]

Thank you for submitting your work entitled "RNA polymerase errors cause splicing defects and can be regulated by differential expression of RNA polymerase subunits" for peer review at *eLife*. Your submission has been evaluated by James Manley (Senior Editor) and three reviewers, one of whom is a member of our Board of Reviewing Editors.

The reviewers have discussed the reviews with one another and the Reviewing Editor has drafted this decision to help you prepare a revised submission.

In this manuscript, Carey describes a computational method called MORPhEUS that allows measuring RNA polymerase error rates using published RNA-seq data. By counting the number of matches and mismatches to the reference genome after technical errors are minimized, MORPhEUS enables the author to estimate the error rate at each position of the genome. Using chromatin-associated RNA from RNA-seq data in K562 cells, the author calculates error rates at splice site junctions and demonstrates that these are higher at 5` splice sites, suggesting that this could affect intron retention. Using ENCODE RNA-seq data, the author further investigates if the expression level of the Pol II subunit Rpb9, known to be required in Pol II fidelity, affects RNA polymerase error rates in different cell lines. Furthermore, the author measures RNA-seq error rates using two yeast strains in which it is possible to modulate the expression of Rpb9 or Dst1, finding that cells with low expression of Rpb9 or Dst1 possess higher error rates, consistent with known biochemical and genetic data. The idea is intriguing and the explanations for observations in terms of Pol II error rates make sense. The proposed method MORPhEUS is appropriate to perform comparative analysis of RNApol induced transcription errors between two or more samples or to identify RNApol errors leading to intron retention. Other than currently available methods this approach is able to identify errors transcriptome-wide and does not "require specialised organism-specific genetic constructs", therefore it seems to be highly useful. The method presented here is interesting and it is a valuable tool for estimating RNA Pol II error rates from RNA-seq data, although several points need to be addressed before publication can be considered.

Essential revisions:

1) Possible alternative explanations for the observations

A main issue with this manuscript is that alternative explanations could also make sense. The author has to show that his explanations are the only or at least most plausible ones. Figure 2 is central to the proposed method. It shows an elevated rate of errors at the uracil in the 5' splice site of the canonical GU-AG introns selected by the author. The explanation given is that Pol II errors in the U lead to intron retention. Why then is the error rate of the guanine not similarly elevated? One would then also expect to see elevated error rates for the conserved AG motif of the 3' splice site and in the well conserved branch point motif. The analysis of these motifs should confirm the interpretation by the author. Because this data is not shown, does that mean no elevated signal has been observed? How can this be explained in the light of the author's interpretation of Pol II errors at splicing motifs leading to retained introns? Since the only position with elevated error rate seems to be the U at the 5' SS, an alternative explanation (probably not the only possible one) could be that some factor strongly binds to the uracil in such a way that the reverse transcription in the RNA-seq protocol causes the uracil to be misread. Note that U->C mutations are also observed in PAR-CLIP and are known to originate during reverse transcription of the RNA.

2) Choice of null model

Figure 2b shows *relative* error rates on the y-axis. The error rates observed around 5' splice sites are normalized by the error rates seen for the same dinucleotides, GT, at other places in the transcriptome. The 4-fold elevated error rate therefore depends on the null model. It would be important to compute the relative error rate at the uracil with more refined trimer null models to see if the 4-fold increase holds up. Two versions, one with the mutated nucleotide at the first position and another model with the mutated nucleotide at the last of the three trimer positions, should be used. The latter version could model sequence-dependent effects during reverse transcription. For each trimer in the transcriptome one can compute the error rate at the first and third nucleotide. Then, the total mutations for each position around the 5' splice site (and the 3' splice site and branch point) are divided by expected numbers of mutations, which is simply the sum of error rates for each of the trimer contexts for the position.

3) Effects of Rpb9

The author demonstrates that expression of Rpb9 negatively correlates with error rates in human cell lines, suggesting that the differential expression of Rpb9 affects RNA polymerase fidelity in vivo. The level of mRNA expression does not necessarily correlate with protein level and, more importantly, the author should normalize the expression of Rpb9 with another subunit of Pol II (e.g. Rpb3) in each cell line used for the analysis (Figure 2). An alternative explanation for Figure 2 and Figure 3 would be that changing Rpb9 and TFIIS concentration from its finely regulated value impairs elongation, which in turn can influence splicing rates and splicing efficiency. (See e.g., Lacadie et al., In vivo commitment to yeast cotranscriptional splicing is sensitive to transcription elongation mutants, Genes Dev. 2006.) Can such alternative explanations be excluded? Further, in Figure 3 the author shows that intron retention is higher under conditions of low Rpb9/Dst1 induction. Is the low induction of Rpb9 or Dst1 affecting the same introns? Does the author find a higher error rate in GT 5´ donor site in the mRNAs that show intron retention?

4) Possible bias resulting from conservation

To measure the error rates at splicing junctions, the author counts errors at each position relative to 5´ donor sites, using reads spanning intron-exon junctions centered on GT donor sites. As a result, the errors at the T nucleotide are more enriched compared to other positions. It is not clear if the analysis is performed measuring the average GT error rate comparing all the reads at intron-exon junctions or single mRNAs (Figure 2). If the analysis is made using all genes, since GT at intron-exon is a conserved sequence and the flanking regions are not, this could lead to a bias. This must be clarified.

5) Suggestions for additional controls

A positive control would be to analyse RNA-seq data of an organism with a mutated polymerase known to have an elevated mutation rate and to show that this mutation rate leads to higher relative error rates at conserved splicing motifs. A negative control would be to analyse RNA-seq data of a mutant organism with a known transcription elongation defect and to show that the elongation defect does not affect the putative Pol II error rate in a similar way as Rbp9 and TFIIs overexpression. If possible we encourage the author to conduct these controls.

6) Repetitive reads

In paragraph four the alignment quality filter procedure is explained. However it is not mentioned how repetitive reads (or potentially repetitive reads in e.g. unknown duplications of genes) are handled and might affect the result. This must be clarified.

7) Possible bias from coverage

Not counting identical mismatches occurring twice or more at the same position (paragraph four) is problematic, because:

– This needs to be adjusted by depth-of-coverage at each position. Positions with high coverage are much more likely to have the same 'real' RNApol error twice, than positions with low coverage. (This seems to be so obvious that we might have overlooked the explanation of the normalization procedure)

– RNA polymerase errors seem to be biased to e.g. C->T (see Figure 3), making it quite a bit more likely to see exactly the same RNApol error twice at a position for C->T/G->A.

In general the uncertainty of RNApol error estimates at low coverage positions (i.e. lowly expressed genes) should be much worse than for high coverage (highly expressed genes). Is this addressed in the algorithm? (Maybe this problem has been discussed but missed by reviewers.) If not it needs some clarification, how different depth-of-coverage and mutation bias is considered when estimating the errors or removing mismatches of the same type.

---

## [Author Response]

1) Possible alternative explanations for the observationsA main issue with this manuscript is that alternative explanations could also make sense. The author has to show that his explanations are the only or at least most plausible ones. Figure 2 is central to the proposed method. It shows an elevated rate of errors at the uracil in the 5' splice site of the canonical GU-AG introns selected by the author. The explanation given is that Pol II errors in the U lead to intron retention. Why then is the error rate of the guanine not similarly elevated? One would then also expect to see elevated error rates for the conserved AG motif of the 3' splice site and in the well conserved branch point motif. The analysis of these motifs should confirm the interpretation by the author. Because this data is not shown, does that mean no elevated signal has been observed? How can this be explained in the light of the author's interpretation of Pol II errors at splicing motifs leading to retained introns? Since the only position with elevated error rate seems to be the U at the 5' SS, an alternative explanation (probably not the only possible one) could be that some factor strongly binds to the uracil in such a way that the reverse transcription in the RNA-seq protocol causes the uracil to be misread. Note that U->C mutations are also observed in PAR-CLIP and are known to originate during reverse transcription of the RNA.

I agree that is was strange that only mutations at the U were enriched in reads spanning intron-exon junctions. Using a newer dataset with a far lower overall mismatch frequency, I find that both the G and U in the 5’ site, and the A in the 3’ site, have higher observed mismatch rates in exon-intron spanning reads (Figure 2 and Figure 2—figure supplement 1).

*2) Choice of null modelFigure 2b shows* relative *error rates on the y-axis. The error rates observed around 5' splice sites are normalized by the error rates seen for the same dinucleotides, GT, at other places in the transcriptome. The 4-fold elevated error rate therefore depends on the null model. It would be important to compute the relative error rate at the uracil with more refined trimer null models to see if the 4-fold increase holds up. Two versions, one with the mutated nucleotide at the first position and another model with the mutated nucleotide at the last of the three trimer positions, should be used. The latter version could model sequence-dependent effects during reverse transcription. For each trimer in the transcriptome one can compute the error rate at the first and third nucleotide. Then, the total mutations for each position around the 5' splice site (and the 3' splice site and branch point) are divided by expected numbers of mutations, which is simply the sum of error rates for each of the trimer contexts for the position.*

Figure 2—figure supplement 1 shows the 3mer error rates for each base in each 3mer. Using this new experiment with higher quality RNA-seq data, only highly elevated error rates are on both the C and G in CG dinucleotides. This suggests that the elevated error rates at the donor and acceptor sites are biological signal and not technical error.

3) Effects of Rpb9The author demonstrates that expression of Rpb9 negatively correlates with error rates in human cell lines, suggesting that the differential expression of Rpb9 affects RNA polymerase fidelity in vivo. The level of mRNA expression does not necessarily correlate with protein level and, more importantly, the author should normalize the expression of Rpb9 with another subunit of Pol II (e.g. Rpb3) in each cell line used for the analysis (Figure 2).

I agree that RNA levels do not necessarily determine protein levels. This is a common caveat with interpreting RNA-seq results. In addition, RNA polymerase complex assembly is highly regulated; knowing the cellular concentration of a specific subunit doesn’t tell you about its phosphorylation status nor how much of that subunit is incorporated into polymerase. I have added text regarding this to the Discussion.

The normalization by the expression of other subunits is a good idea. I have added a figure (Figure 2—figure supplement 2) showing that RPB9 expression negatively correlates with RNA-seq mismatch rates when normalized by either RPB3 or by the median expression of all subunits.

An alternative explanation for Figure 2 and Figure 3 would be that changing Rpb9 and TFIIS concentration from its finely regulated value impairs elongation, which in turn can influence splicing rates and splicing efficiency. (See e.g., Lacadie et al., In vivo commitment to yeast cotranscriptional splicing is sensitive to transcription elongation mutants, Genes Dev. 2006.) Can such alternative explanations be excluded?

I cannot think of a good experiment to determine if the difference in splicing due to RBP9 and DST1 underexpression (Figure 3) are due to a change in elongation rates, error rates, or both. I believe that the new data showing elevated mismatch frequencies at both 5’ and 3’ splice sites lends further support to RNA polymerase errors being responsible for at least some of the difference in splicing.

Further, in Figure 3 the author shows that intron retention is higher under conditions of low Rpb9/Dst1 induction. Is the low induction of Rpb9 or Dst1 affecting the same introns? Does the author find a higher error rate in GT 5´ donor site in the mRNAs that show intron retention?

Unfortunately, because the error rates are lower than the coverage at any one position, the error rate at any particular exon is dominated by sampling bias. We are in the process of developing combining single-molecule unique IDs with targeted sequencing to ask this very question, but cannot do so using standard RNA-sequencing data.*4) Possible bias resulting from conservationTo measure the error rates at splicing junctions, the author counts errors at each position relative to 5´ donor sites, using reads spanning intron-exon junctions centered on GT donor sites. As a result, the errors at the T nucleotide are more enriched compared to other positions. It is not clear if the analysis is performed measuring the average GT error rate comparing all the reads at intron-exon junctions or single mRNAs (Figure 2). If the analysis is made using all genes, since GT at intron-exon is a conserved sequence and the flanking regions are not, this could lead to a bias. This must be clarified.*

The error rates at exon-intron junctions (Figure 2) are compared to error rates within exons (Figure 2—figure supplement 1). I agree that this was not clear, and have clarified it in the Materials and methods section.*5) Suggestions for additional controlsA positive control would be to analyse RNA-seq data of an organism with a mutated polymerase known to have an elevated mutation rate and to show that this mutation rate leads to higher relative error rates at conserved splicing motifs.*

I don’t have reason to believe that an RNA polymerase fidelity mutant will have a larger increase in error rates at conserved splicing motifs relative to the increase at other positions. In the mutants, the increase in error rates at splice junctions should be the same as at other similar sequences context. The greater error rate at splice junctions is because these errors can result in intron retention.

A negative control would be to analyse RNA-seq data of a mutant organism with a known transcription elongation defect and to show that the elongation defect does not affect the putative Pol II error rate in a similar way as Rbp9 and TFIIs overexpression. If possible we encourage the author to conduct these controls.

This was a very nice suggestion. I’ve done it (Figure 3—figure supplement 1).

To determine if defects in elongation result in increased RNA-seq mismatch frequencies, I analyzed RNA-seq data from spt4 and elc1 strains, which as shown in Lacadie et al., do not have fidelity defects. I see no difference in RNA-seq mismatch frequencies, suggesting that perturbations that affect elongation would not results in an increase in RNA-seq mismatches.*6) Repetitive readsIn paragraph four the alignment quality filter procedure is explained. However it is not mentioned how repetitive reads (or potentially repetitive reads in e.g. unknown duplications of genes) are handled and might affect the result. This must be clarified.*

Reads that map to multiple locations in the genome are discarded. I’ve clarified this in the text.*7) Possible bias from coverageNot counting identical mismatches occurring twice or more at the same position (paragraph four) is problematic, because:– This needs to be adjusted by depth-of-coverage at each position. Positions with high coverage are much more likely to have the same 'real' RNApol error twice, than positions with low coverage. (This seems to be so obvious that we might have overlooked the explanation of the normalization procedure)– RNA polymerase errors seem to be biased to e.g. C->T (see Figure 3), making it quite a bit more likely to see exactly the same RNApol error twice at a position for C->T/G->A.In general the uncertainty of RNApol error estimates at low coverage positions (i.e. lowly expressed genes) should be much worse than for high coverage (highly expressed genes). Is this addressed in the algorithm? (Maybe this problem has been discussed but missed by reviewers.) If not it needs some clarification, how different depth-of-coverage and mutation bias is considered when estimating the errors or removing mismatches of the same type.*

I have added a supplementary figure showing both (1) this filtering does not affect the results, and (2) the statistical reasoning for this filter (Figure 1—figure supplement 2). Briefly, the error rate of RNA polymerase is on the order of 10^-5^, while 90% of positions have a coverage of <10^2^. Therefore, while many positions in the genome exhibit multiple identical errors the likelihood of observing multiple identical errors is very low.